# Submicronic-Scale Mechanochemical Characterization of Oxygen-Enriched Materials

**DOI:** 10.3390/nano14070628

**Published:** 2024-04-03

**Authors:** Marie Garnier, Eric Lesniewska, Virgil Optasanu, Bruno Guelorget, Pascal Berger, Luc Lavisse, Manuel François, Irma Custovic, Nicolas Pocholle, Eric Bourillot

**Affiliations:** 1Laboratory Interdisciplinaire Carnot de Bourgogne (ICB), UMR 6303 CNRS, University of Bourgogne, 21000 Dijon, France; 2Laboratory of Mechanical & Material Engineering (UR LASMIS), University of Technology Troyes, 10300 Troyes, France; bruno.guelorget@utt.fr (B.G.);; 3Laboratory Nanoscience and Innovation for Materials, Biomedecine and Energy (NIMBE), UMR 3685 CEA-CNRS, University of Paris-Saclay, 91191 Gif-sur-Yvette, France

**Keywords:** light chemical element quantification, diffusion, local mechanical properties, scanning microwave microscopy, oxide-metal interface materials

## Abstract

Conventional techniques that measure the concentration of light elements in metallic materials lack high-resolution performance due to their intrinsic limitation of sensitivity. In that context, scanning microwave microscopy has the potential to significantly enhance the quantification of element distribution due to its ability to perform a tomographic investigation of the sample. Scanning microwave microscopy associates the local electromagnetic measurement and the nanoscale resolution of an atomic force microscope. This technique allows the simultaneous characterization of oxygen concentration as well as local mechanical properties by microwave phase shift and amplitude signal, respectively. The technique was calibrated by comparison with nuclear reaction analysis and nanoindentation measurement. We demonstrated the reliability of the scanning microwave technique by studying thin oxygen-enriched layers on a Ti-6Al-4V alloy. This innovative approach opens novel possibilities for the indirect quantification of light chemical element diffusion in metallic materials. This technique is applicable to the control and optimization of industrial processes.

## 1. Introduction

A major problem in the metallurgy, aeronautics, automotive, energy, space and nuclear industries is the ability of metal parts to tolerate ‘extreme’ conditions that have an impact on their lifespan. Indeed, under severe conditions such as high temperature and pressure, the integrity of a system can be altered by the diffusion of light chemical elements with a loss of ductility and an increase in corrosion [1,2]. We have focused our research on titanium and titanium alloys, which are widely used in many industrial sectors, including healthcare.

Titanium and its alloys are widely used due to their excellent physico-chemical properties: low density, high specific strength, high fatigue strength [3], excellent corrosion and oxidation resistance and biocompatibility. As an allotropic material, titanium has two crystal structures, depending on temperature and the composition of any alloying elements. At high temperatures, pure titanium exhibits a centered cubic β-phase crystallographic lattice (bcc), while at lower temperatures, it reversibly transitions to a compact hexagonal α-phase (hcp). This transition takes place for pure Ti at 882 °C, which is called the transition temperature β [4,5]. In addition, the characteristics of titanium can be modified in a controlled way by introducing various alloying elements. By adjusting the choice and proportion of these alloying elements, it is possible to impact the amount and distribution of β and α phases in the material’s microstructure, as well the mechanical properties at low and high temperature [4,6]. As such, titanium alloys can adapt different phase combinations (alpha and beta or the combination of both, alpha plus beta), thus enlarging the possibilities of tailoring material properties in order to obtain specific application needs [5,6]. While the mechanical properties of pure titanium (cp-Ti) are relatively modest, its various alloys, such as Ti-6Al-4V or Ti-6Al-7Nb, offer superior levels of strength, hardness and toughness, thus meeting higher requirements in a diverse range of applications [4,5,6].

Titanium’s excellent ability to resist corrosion is based primarily on its high affinity with oxygen. This affinity induces the spontaneous formation of a passive, self-healing protective layer on the surface of the material, mainly composed of TiO_2_ [4,7]. Previous studies have revealed that different alloying elements can influence this oxide layer in several ways, including its composition, density and stability [8]. It is therefore crucial to characterize this oxide layer and the diffusion of oxygen in the material, which is inherent to high-temperature conditions, and finally, to understand the impact of oxygen diffusion on the material’s mechanical properties. The local nanomechanical properties [9] could significantly impact mesoscopic material properties such as conductivity and corrosivity in the case of microstructures, among others [10].

It is well known that oxygen contents superior to 1 at. % lead to a dramatic loss of ductility and then risks of failure by fatigue [11,12]. The characterization of the oxygen-enriched zone can be done using conventional techniques such as micro-hardness testing, nanoindentation, energy dispersive X-ray spectroscopy (EDS) and nuclear reaction analysis (NRA) [13]. However, these techniques are constrained due to the low spatial-resolution performance set by their intrinsic technical properties. Micro hardness testing is widely used to measure the hardness value of small regions according to the micro indentation hardness principle. Although widely used to determine a material’s hardness or resistance to penetration, micro hardness testing is unable to retrieve mechanical properties at nanoscale area of oxygen enrichment [14]. EDS techniques can promote constraints in their applications since the preparation of cross-section of samples is inevitably accompanied with the contamination of the superficial layers by the atmospheric oxygen or the oxygen contained in the polishing products, which makes the precise quantification of oxygen hazardous. NRA is a powerful technique but requires heavy equipment and complex experimental procedures.

Given the recent developments of another atomic force microscopy (AFM) technique called scanning microwave microscopy (SMM), it has been demonstrated that it is possible to accurately quantify the distribution of light elements in metallic materials by exploiting the microwave phase signals of SMM. Scanning microwave microscopy (SMM) is a microwave-based probing method integrated into the AFM platform that enables the tomographic mapping of electromagnetic properties with the nanoscale resolution defined by the size of AFM cantilever. It can be used to investigate dielectric and conductive properties [15,16,17,18,19].

In our recent studies, we have detected subsurface defects and a gradient of properties produced by the presence of dissolved oxygen in the material by scanning microwave microscopy [20,21,22]. We obtained a proportional SMM phase shift response to the oxygen concentration by acquiring SMM measurements at different frequencies (0.3–18 GHz). In addition, specific microwave frequencies allowed us to obtain the cartography of dissolved oxygen in the metal lattice as a function of depth under the scanned surface [20]. Lately, a similar study has been carried out on laser impact texturing, which showed that the analysis of the SMM amplitude signal shift was related to the mechanical properties induced by laser impacts [23].

In this article, we propose an innovative approach for the high-resolution characterization of oxygen enrichment in a Ti-6Al-4V alloy by examining the SMM amplitude and phase shift obtained in the microwave range of 0.3–18 GHz. In addition, we compared the SMM analysis with conventional techniques such as NRA and nanoindentation.

SMM amplitude and phase signals provide two simultaneous types of information corresponding to those obtained by conventional techniques (NRA and nanoindentation) while offering greater precision and a better resolution. This innovative approach has a significant impact on the assessment and control of light scattering phenomena in metallic materials, with the aim of effectively optimizing material properties and requirements for a variety of applications.

## 2. Theoretical Approach

The scanning microwave microscopy was initially developed for calibrating doping measurements in semiconductors [24,25,26,27,28] while the latest applications were carried out on metals and their alloys [20,21,22,29].

The scanning microwave microscopy (SMM) combines the functionality of an atomic force microscope (AFM) tip with a microwave vector network analyzer (VNA). It allows the simultaneous measurement of topographic and electromagnetic properties of a sample over a wide frequency range 0.3 to 18 GHz. Figure 1 presents a schematic representation of the AFM coupled with a resonant circuit and a VNA.

The microwave signal is sent directly from the vector network analyzer and transmitted through a resonant circuit to a conductive probe in contact with the sample. A half-wavelength impedance transformer is placed directly on a 50 Ω load to form a matching resonant circuit.

The complex reflection coefficient is measured directly from the VNA. It corresponds to the reflected signal on the incident signal. Microwave measurement is obtained by measuring the complex reflection coefficient from the network analyzer:(1)$$\begin{array}{c} S_{11}=20\log\left|\Gamma \right|\;\;with\;\Gamma =\frac{\left(Z-1\right)}{\left(Z+1\right)}=\frac{V_{ref}}{V_{inc}\;} \end{array}$$
where Γ is the complex reflection, *Z* is the normalized impedance, *V_ref_* is the reflected signal and *V_inc_* is the incident signal, respectively.

The measured impedance *S*_11_ presents the local impedance, which results from the tip–sample interaction and simultaneously provides the amplitude and phase of *S*_11_ signal with topography. The amplitude measured depends on the variation in resistivity and therefore the conductivity of the material.

The phase corresponds to the dielectric losses in the material directly related to the composition of the material [15].

In the scanning microwave microscope, the contact mode is often used with a minimal interaction force, typically less than 750 pN. The AFM tip also acts as a radiating local antenna, transmitting and receiving the microwave signal at the point of contact. The tip–sample junction acts as a near-field coupling element, enabling interaction between the microwave field and the electromagnetic properties of the sample in the immediate vicinity. By measuring the microwave signals reflected at the point of contact, we can extract local electromagnetic information from the sample. This allows us to characterize the electrical properties of the sample with nanoscale spatial resolution in addition to topographical data.

The leverage of SMM relies in the possibility of obtaining amplitude and phase images at different microwave frequencies [22], which allows the establishment of in-depth cartography based on skin effect:(2)$$\begin{array}{c} \delta =\frac{1}{\sqrt{\pi \mu_{0}\mu_{r}\sigma f}} \end{array}$$
where *δ* is the microwave penetration depth (skin effect), *μ*_0_ is the magnetic permeability of the vacuum, *μ_r_* is the relative permeability of the metal, *σ* = 1/ρ is the conductivity of the metal (ρ is the resistivity of the metal) and *f* is the scanning frequency.

The penetration of microwaves into the sample, which depends on used frequency, makes it possible to identify the various factors likely to alter conductivity, such as the diffusion of light elements, defects, phase transition or mechanical stress. Figure 2 shows the zoomed insight of the absorption spectrum peak of amplitude and phase signal. Resonance frequencies correspond to the frequencies at which the peaks in the absorption spectrum are located. Any change in the impedance circuit has an influence on the shape and position of the peak, as well as on the phase value. Near the position of a resonant frequency peak (Figure 2), the spectrum presents the change in the impedance of the sample (outline by green) compared with the reference spectrum (outline by blue). By recording the amplitude and phase of the spectrum at a given frequency at any point on the scanned surface, it is possible to obtain local information about the sample. During the sweep, the amplitude and phase are recorded at a selected frequency (close to the resonance frequency). The dotted red line in Figure 2 represents the fixed frequency chosen for the scan. The value of the microwave frequency is chosen in a region around the resonant frequency where the phase has a linear variation in order to obtain the most accurate results.

## 3. Experiments and Results

### 3.1. Preliminary Study on Pure Zirconium

Thus, the coefficient *S*_11_, the complex reflection coefficient, simultaneously gives the amplitude and the phase of the microwave signal, respectively, related to the material analyzed by its resistive component (or conductivity of the material) and its reactive component (or composition via the dielectric losses of the material). To asses the reliability and sensitivity of scanning microwave microscopy (SMM), we recently correlated amplitude and phase signals to residual stress induced by shot peening on a zirconium metal surface [30]. In this study, shot peening was carried out on rectangular zirconium plates measuring 80 × 40 × 2 mm^3^ for 10 min for a first sample and 30 min for a second. The parameters were chosen to induce residual stresses within a layer approximately 400 μm thick beneath the metal surface. This method involves the use of tungsten carbide balls measuring 2 mm in diameter and a vibrating sonotrode operating at 20 kHz with an amplitude of 24 μm. To characterize the residual stresses in the material, the incremental hole-drilling method was used. Details about the general method can be found in the ASTM E837 standard [31]. This method consists of bonding a strain gauge rosette to the tested material and drilling a hole in small increments in a precise place next to the strain gauges. After each increment, the relaxation strain is measured by the strain gauges. The residual stresses are then calculated from the measured strains around the hole using suitable calculated coefficients, strain gauge rosette characteristics and material parameters.

In Figure 3, residual stress profiles are depicted for samples subjected to shot peening for durations of 10 and 30 min. The measurements reveal a distinctive stress distribution pattern: the outermost layers experience slight compression, and as one goes deeper, a negative stress gradient reaches a minimum algebraic value at a depth of approximately 110 μm (dashed line) for 10 min of shot peening and 180 μm (dashed line) for 30 min. Subsequently, there is an inversion of the gradient sign, leading to positive values (tensile stress). This particular profile is a well-documented phenomenon in the literature, as is the fact that the compression peak is offset from the surface and increases in amplitude with shot peening time.

Since SMM imaging is not essential in this type of study and given the spatial distribution of the effect of macroscopic residual stresses, we have streamlined the SMM analysis method to improve efficiency. Instead of capturing a series of images, we generated an amplitude spectrum for each measurement point (spaced 50 μm apart in this case), using the movement of the motorized stage to obtain measurement steps ranging from 5 μm to a few millimeters. Subsequently, at a constant frequency (in this case, 9.6 GHz), we graphed the amplitude variation in relation to the distance; thus, this signal has been directly correlated to the elasto-resistive properties of the material.

Consequently, measurements of the shot-peened Zr samples were carried out as described above. The shot peening profiles can be observed in Figure 3. In order to efficiently discern the compression part of the tension part for each of the profiles, the average value of the microwave amplitude signal profile is subtracted at each measurement point to obtain a signal of 0 dB. This shows the excellent correlation between the profiles obtained by SMM and those obtained by indentation, with the curves showing the compression and tension zones perfectly.

It is interesting to note that the SMM gives the residual stress profile directly, without intermediate calculation, simply by comparing the amplitude of the microwave signal at the same frequency at each measurement point.

Another application of scanning microwave microscopy is the detection of light chemical elements such as oxygen [20,21,22]. To achieve this, a pure zirconium plate (99.2% Zr from Goodfellow, Huntingdon, UK) was oxidized. It was first annealed at 750 °C under secondary vacuum (1 × 10^−6^ bar) for 2 h, then oxidized in air under atmospheric pressure at 650 °C for 72 h. After oxidation, the sample was cross-sectioned and mirror-polished, then analyzed by nuclear reaction analysis (NRA) to determine the %at of oxygen that had diffused into the sample and finally compared with the results obtained with the SMM technique.

The results were presented in a previous article by E. Bourillot et al. [20]. The topography image clearly revealed the grains of the metal. The topographical profile indicated that the ZrO_2_ region exhibits a distinct surface roughness compared to the Zr–O and Zr zones. The roughness amplitude measures around 20 nm, which is notably small when considering the size of the scanned zone, which is 80 μm. The SMM phase shift image clearly showed variations in the phase signal, which is introduced solely by a composition change in the material. Only the dissolution of oxygen in the zirconium lattice produces these compositional variations. The proof is that in an oxygen-free zone, the Zr pure area, the microwave phase signal shows no variation. To further assess the sensitivity of the SMM to oxygen detection, the microwave phase signal was compared with the result obtained by NRA at the oxide-metal interface. The results unambiguously showed a proportional relationship between the SMM phase shift and the oxygen concentration obtained with the NRA measurement.

These results show that the microwave amplitude signal is related to the mechanical state of the material, and the phase signal to the chemical species in the material. In the present paper, the aim is therefore to see simultaneously and separately how the mechanical state of a material is affected by the scattering of light chemical elements as a function of this scattering.

### 3.2. Simultaneous Detection of Amplitude and Phase Signal on Titanium Alloy

The titanium alloy used in this study is the common Ti-6Al-4V (Timet, nominal composition in weight%: Al 6.23%, V 3.95%, Fe 0.14%, O 0.17%, Ti bal.). This alloy is widely used in the industry, particularly for aerospace applications or medical use (implants, surgery) [32,33]. The sample was cut from a 1.6 mm thick plate into an 8 × 10 mm^2^ rectangle and then oxidized under controlled air flow in a thermo-gravimetric analyzer at 750 °C. After oxidation, the sample was cut and mounted in a hard and rigid resin, and then the cross section was mirror polished. The polishing was done with particular care to produce a surface as flat as possible.

The Ti-6Al-4V alloy was chosen for the anionic nature of its high temperature oxidation. The O_2_-ions diffuse through the oxide via the oxygen vacancies, and some of them produce new oxides at the interface metal/oxide while the others continue their diffusion as O atoms inserted into the octahedral sites of the lattice of titanium to form a solid solution, also called alpha-case. The diffusion profile extends, in our case, to several tens of micrometers deep under the oxide scale. At the metal/oxide interface, the oxygen concentration value can reach high values, up to 33 at. %. In general, the oxygen concentration profile is close to an erfc type function as a function of the depth [34]. The diffusion depth can go up to several hundred micrometers depending on the exposure time and temperature. The main parameter influencing the diffusion depth is the diffusion coefficient of oxygen in the metal at this temperature.

All measurements (SMM, NRA, nanoindentation) are performed in am oxygen-enriched solid solution. Figure 4 shows the location of the measurement zone. It is located in the metal after the metal-oxide interface.

#### 3.2.1. NRA Measurements

The nuclear reaction analysis was conducted utilizing a Van der Graaff linear accelerator with a maximum energy of 3.7 MeV. This nuclear microanalysis relies on the interactions of energetic deuterons (specifically 2H^+^, in the MeV range) with the material under examination. The concentrations of constituent elements were determined through spectroscopy after the emitted particles induced nuclear reactions (NRA, nuclear reaction analysis), resulting in the emission of various particles such as protons, alpha particles, X-rays and gamma photons (known as particle-induced X-ray emission, PIXE, or particle-induced gamma-ray emission, PIGE) or elastically scattered particles (RBS, Rutherford back scattering). In this experiment, deuterons with an energy of 1.45 MeV were used. NRA is particularly suited for the analysis of light elements and their isotopes. For example, the analysis of oxygen-16 involves the spectroscopy of protons produced by the interaction of a deuteron beam with oxygen nuclei, resulting in the reaction 16O(d,p1)17O [35].

In this study, only oxygen concentrations below 30 at. % were considered in order to avoid regions that could partially contain oxides. The size of the spot used for NRA measurements was 3.5 × 2.5 µm^2^. Thus, below 30 at. % oxygen content, the spot probed only the solid solution without any risk to overlap any oxide zone. For quantification purposes, standards such as SiO_2_, TiN and C were used to calibrate the measurements [36]. The oxygen profile obtained by NRA is shown in Figure 5.

#### 3.2.2. SMM Amplitude and Phase Measurements

Microwave microscopy measurements were made with a vector network analyzer (VNA N5230A, Agilent Technology, Santa Clara, CA, USA) and an atomic force microscope (AFM 5600LS, Agilent Technology) [20,21,22,29]. The probes used in this study were AFM RMN-25PT400B probes with a nominal spring constant of about 10 N/m (Rocky Mountain Nanotechnology, Salt Lake City, UT, USA). Full amplitude spectra were recorded in static. In imaging, a fixed frequency is required to scan the area to be studied. A frequency is chosen according to the modalities explained above. Since the frequency determines the depth of microwave penetration, it can also be chosen according to the desired depth. Once the frequency was chosen, an area of 80 × 80 µm^2^ was scanned and the topography (Figure 6a), the amplitude of the reflected microwave signal (Figure 6b) as well as the phase shift between the incident and reflected wave (Figure 6c) were recorded.

The frequency used for SMM analyze (below 7.57 GHz) corresponds to frequency at the penetration depth δ ≈ 7.56 µm where the profiles were not affected by polishing.

All these data were stored as images. For an 80 × 80 µm^2^ image, the measurement step is 150 nm while the acquisition rate is 10 min. The large scan profile, shown in Figure 6d,e, was obtained by scanning two adjacent zones with the size of 80 µm, each. Graphical amplitude and phase profiles were extracted from five lines on the amplitude and phase images. A study of the repeatability of measurements within a single image was thus established. Consequently, each profile presented in this study represents the average variation in phase shift (Figure 6e) and amplitude (Figure 6d) caused by the insertion of oxygen into the metal from several cross-sections obtained from microwave images.

The SMM measurement was carried out in the center of the sample (green outline in Figure 7). It can be seen that the topographies in Figure 6 and Figure 7 are similar in the metal zone. The phase and amplitude images in Figure 7 show that the information obtained by the SMM measurement is not related to the topography of the sample.

#### 3.2.3. Nanoindentation Measurements

Instrumented nanoindentation is a widely used technique to measure Young’s modulus and hardness [37,38,39]. A pyramidal diamond Berkovich tip calibrated on SiO_2_ is pressed into the sample while continuously measuring the load and the penetration depth (Figure 8a). The continuous stiffness measurement mode [40,41] allows constant measurement of Young’s modulus and hardness during the penetration process based to the following equations:(3)$$\begin{array}{c} E_{r}=\frac{\sqrt{\pi }}{2\beta }\frac{S}{\sqrt{A_{c}}}\;\;\;\;\;and\;\;\;\;\;H=\frac{P}{A_{c}}\; \end{array}$$
where *E_r_* is the reduced modulus, *β* presents a constant with value 1.034, *S* is the contact stiffness, *A_c_* is the projected contact area, *H* presents the hardness and *P* presents the load.

Therefore, Young’s modulus can be easily deduced from the reduced modulus:(4)$$\frac{1}{E_{r}}=\frac{\left(1-\nu^{2}\right)}{E}+\frac{\left(1-{\nu_{i}}^{2}\right)}{E_{i}}\;$$
where *E* and *ν* are the Young’s modulus and Poisson’s ratio, respectively, for the specimen and *E_i_* and *ν_i_* present same parameters for the indenter.

The indent was 800 nm deep. The modulus and hardness values given in Figure 8b were the average of measurements during the loading process between 400 and 780 nm. In CSM mode, five indentations were averaged to determine average hardness and Young’s modulus values for statistical purposes.

In Figure 8, the nano hardness and Young’s modulus have a similar profile with a similar decrease. In Figure 8, there is a linear relationship between them in the oxygen-enriched solid solution. In this study, we will only deal with the Young’s modulus results.

## 4. Discussion

The measurements were performed in the cross sections of the oxidized plate by NRA, nanoindentation and microwave techniques.

This provided information on the chemical (at. % oxygen) and physical (Young’s modulus and nano hardness) characterization. The microwave results can thus be correlated with the results of the other methods.

### 4.1. Chemical and Physical Characterization by NRA and Nanoindentation

Young’s modulus and oxygen concentration have been shown to exhibit a non-linear relationship [42]. The comparison between Young’s modulus and oxygen concentration in the enriched zone is shown in Figure 9. In Figure 9, the variation in Young’s modulus with depth is not the same as the variation in oxygen concentration. In Figure 9, these two empirical signals are not linearly related.

### 4.2. SMM Characterization of Oxygen-Enriched Zone

In this study, we focused on the oxygen-enriched sublayer, which has a different electrical conductivity to the initial material. Figure 10 highlights the difference in variation between the two SMM signals: amplitude and phase.

The SMM measurement therefore provides two signals whose variation is non-linear. These two measurements are compared to the oxygen content and the Young’s modulus to highlight the relations between these different signals.

### 4.3. Comparison of the Oxygen Concentration and SMM Signals

Figure 11a,b compare the SMM signals with an NRA measurement. Figure 11a shows the comparison between the NRA measurements and the microwave phase shift in the oxygen-enriched region. The decrease of the two profiles seems to be concordant. The differences can be explained by the fact that the SMM measurements were not performed exactly at the same location as the NRA quantification. There may be slight local differences between several oxygen profiles because the alloying elements may not be homogeneously distributed. It may also be noted that the variation in the SMM signal is less pronounced near the interface. This is due to the much higher spatial resolution of SMM compared with NRA. Indeed, the NRA spot size is 3.5 × 2.5 µm^2^, and the step size is 1 µm. As a result, the volume studied in NRA measurements is much larger than that of SMM measurements. In the case of the SMM, the measurement line is extracted from a cross-section made on image of 80 µm × 160 µm each consisting of 512 points × 1024 points, with a measurement step every 156 nm, and is hence a better resolution for SMM measurements. To compare the two results, SMM and NRA, a point was taken every 1 µm on the SMM measurements to compare it with the corresponding measurement point in NRA. This could explain the slight shifts mentioned between the two measurements. Thus, the localization of the metal/oxide interface is much more accurate, which is a strong advantage of the scanning microwave technique.

We can notice that the relation between oxygen content and microwave phase shift seems to be linear, with the equation:(5)$$\begin{array}{c} at.\%\;O=35.66\times \varphi +0.2235 \end{array}$$
and an R2 (determination coefficient) of 0.96. This calibration function allows us to evaluate the dissolved oxygen concentration in the material directly from the microwave phase shift measurements. Figure 11b shows the comparison between the oxygen concentration obtained by NRA measurements and the microwave amplitude signal in the oxygen-enriched zone. The decrease of the two profiles is not linear. This means that the variation oxygen concentration measured by NRA does not have the same dependence law as the amplitude versus the depth. The relation is therefore not linear. This seems to be coherent since the phase and amplitude do not follow the same law and are not linear.

### 4.4. Comparison of Young’s Modulus and SMM Signals

Microwave measurements were also compared with nanoindentation measurements. The Young’s modulus evolution measured by nanoindentation in the oxygen solid solution was compared with the amplitude signal measured by scanning microwave microscopy, as shown in Figure 12a. This comparison highlights a linearity between the measurements of the two techniques. Empirically, there is a linear dependence between the microwave amplitude signal and the Young’s modulus by nanoindentation measurement. The microwave amplitude signal would therefore allow us to obtain by comparison the Young’s modulus which is a mechanical property of the material.

Since phase and amplitude do not have the same linear dependence, and amplitude has the same dependence as Young’s modulus, it would be consistent for phase shift and Young’s modulus not to follow the same profile. Young’s modulus and phase are compared to verify this hypothesis in Figure 12b. It shows that there is no linearity between microwave phase shift and Young’s modulus.

## 5. Conclusions

In this paper, we show that scanning microwave microscopy (SMM) is a powerful technique for identifying enriched subsurface zones and locating the oxide/metal interface with nanoscale resolution. A linear dependence is observed between the microwave amplitude signal and the elastic modulus, indicating a correlation between the electrical and mechanical properties of the sample. This correlation can provide valuable information on material behavior and characteristics. A linear dependence is also observed between microwave phase shift and oxygen concentration, suggesting that oxygen-induced changes in electrical properties in the material can be detected using microwave microscopy. By taking advantage of this linear relationship, microwave microscopy can indirectly quantify the concentration of dissolved oxygen in the material. This can be particularly useful for studying the length of diffusion in oxygen-sensitive materials. In addition, it could also be useful in the study of semiconductors, catalysts or biological samples, where oxygen-induced electrical changes play an important role in their behavior or performance. By integrating microwave microscopy results with complementary techniques, researchers can validate their findings, explore underlying mechanisms and gain a more complete understanding of sample properties, ultimately improving overall data analysis and interpretation. The perspective of our studies is to enlarge the application of SMM as unique technique for submicronic mechanochemical characterization. As such, we aim to calibrate SMM amplitude and phase measurements for diverse types of metallic materials with different oxidation levels.

## Figures and Tables

**Figure 1 nanomaterials-14-00628-f001:**
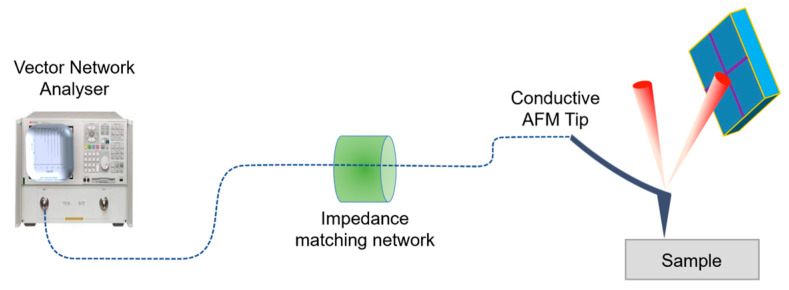
Scanning microwave microscope setup.

**Figure 2 nanomaterials-14-00628-f002:**
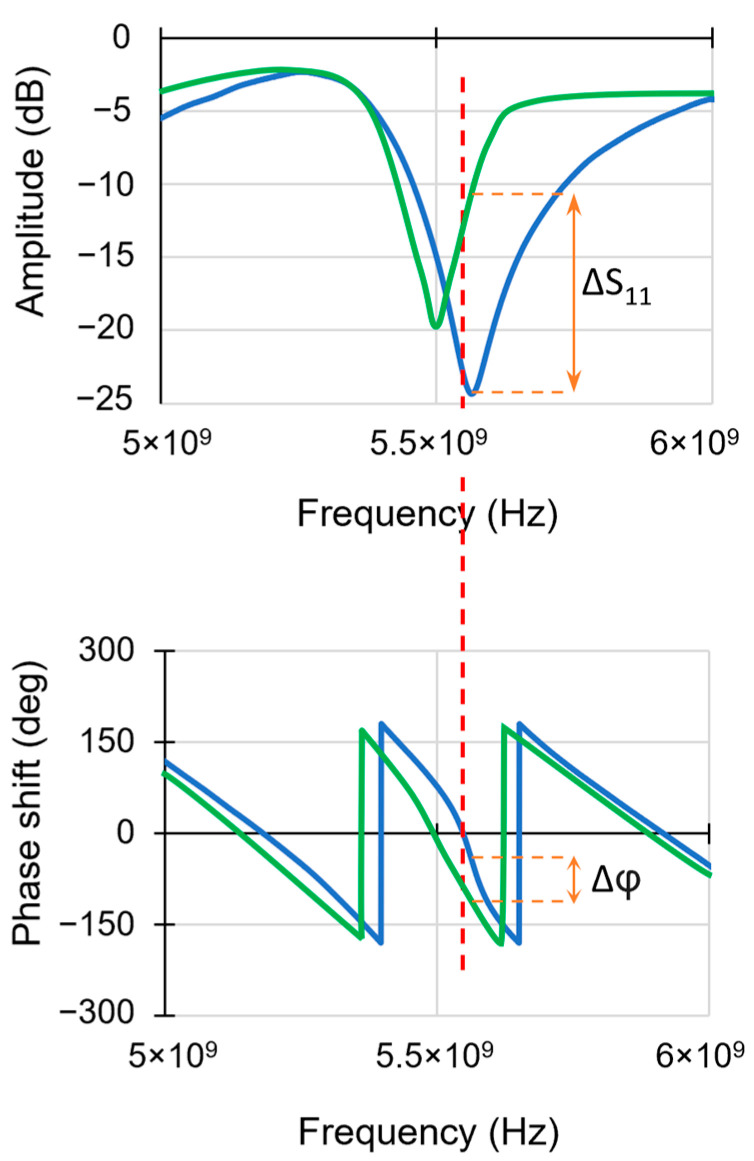
Absorption spectrum peak of amplitude and phase near the position of resonant frequency. The shift between spectrum of reference (outlined by blue) and sample (outlined by green) presents the change of impedance.

**Figure 3 nanomaterials-14-00628-f003:**
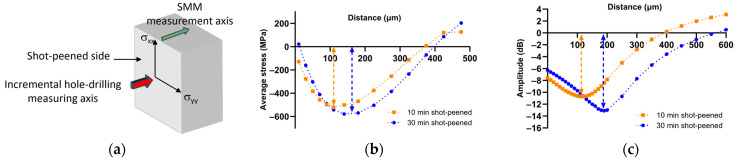
Comparison of residual stress profiles between incremental hole-drilling and SMM measurements. (**a**) Schematic measurement method; (**b**) Residual stress profile of shot peened Zr obtained by incremental hole-drilling method; (**c**) Microwave amplitude signal variation.

**Figure 4 nanomaterials-14-00628-f004:**
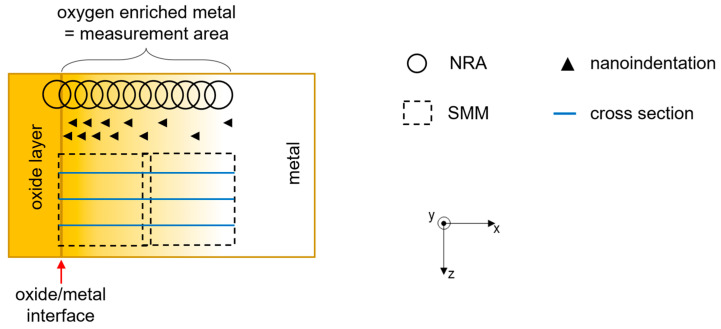
Schema of the different areas of an oxidized metal sample.

**Figure 5 nanomaterials-14-00628-f005:**
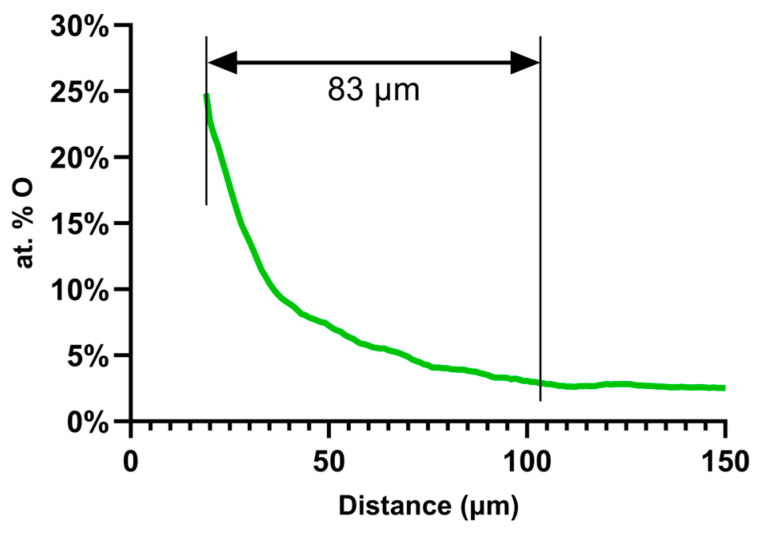
Graphic of atomic percentage (at. %) of oxygen measured by NRA on a TA6V cross-section sample oxidized at 750 °C for 100 h.

**Figure 6 nanomaterials-14-00628-f006:**
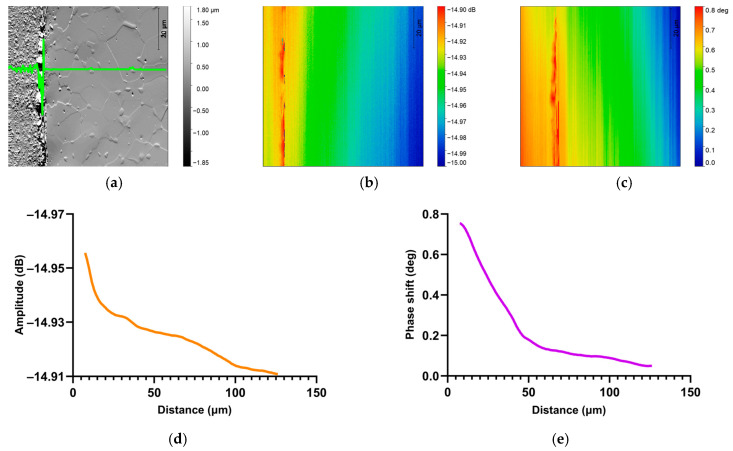
Scanning Microwave Microscopy imaging of TA6V at resonant frequency 7.57 GHz. (**a**) SMM Topography (80 × 80 µm^2^) and corresponding (**b**) amplitude and (**c**) phase images. Graphical representation of average (**d**) amplitude and (**e**) phase values at 7.57 GHz.

**Figure 7 nanomaterials-14-00628-f007:**

SMM imaging of Ti-6Al-4V sample at resonant frequency f = 7.57 GHz. (**a**) SMM topography image (80 × 80 µm^2^) with (meaning of green line) and corresponding (**b**) amplitude and (**c**) phase images.

**Figure 8 nanomaterials-14-00628-f008:**
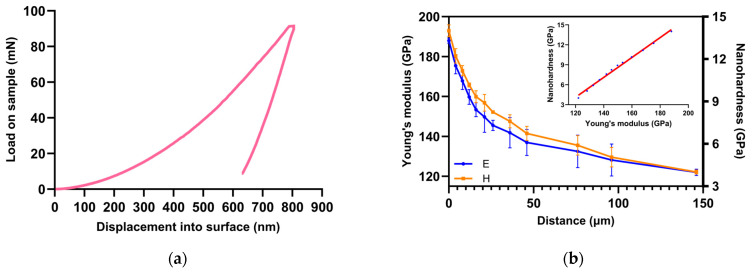
(**a**) Charging and discharging curve of a nanoindentation (**b**) Graphic representation of hardness (outline by orange) and Young’s modulus (outline by blue) of solid solution enriched by oxygen and measured by nanoindentation and calibration function.

**Figure 9 nanomaterials-14-00628-f009:**
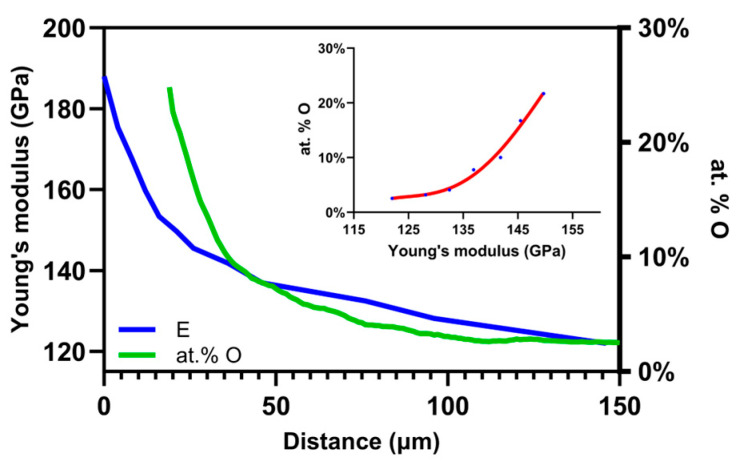
Graphic representation of the comparison between (meaning of at. %) oxygen measured by NRA (outlined by green) and Young’s modulus (outlined by blue) measured by nanoindentation and calibration function, respectively (blue dots red line).

**Figure 10 nanomaterials-14-00628-f010:**
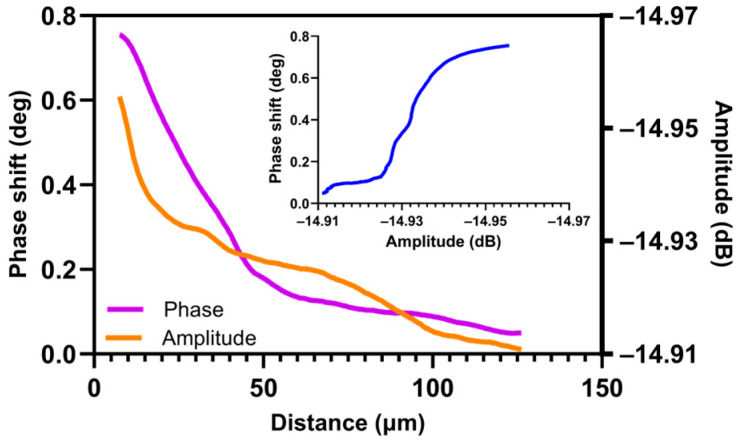
Graphic representation of comparison between phase shift and amplitude profiles measured by SMM at f = 7.57 GHz and calibration function (outline by blue).

**Figure 11 nanomaterials-14-00628-f011:**
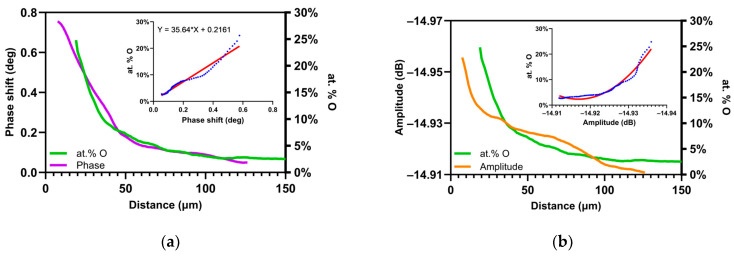
Graphic representation of comparison between oxygen concentration profile for TA6V obtained by NRA (outline by green) and (**a**) phase shift values (outline by purple) and (**b**) amplitude profile (outline by orange). Both graphic representations show similar linear behavior as calibration function (input outline by blue).

**Figure 12 nanomaterials-14-00628-f012:**
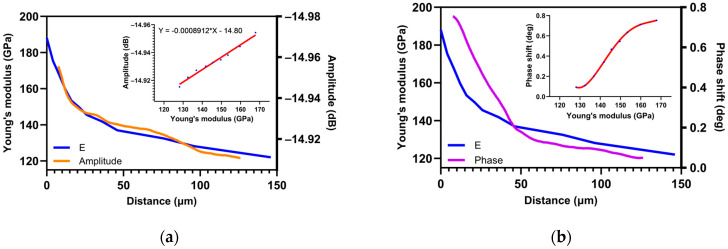
Graphic representation of (**a**) SMM amplitude profile (outline by orange) and Young’s modulus (outline by blue) with an insert of calibration function. (**b**) SMM phase shift (outline by purple) and nanoindentation measurements (Young’s modulus outline by blue) with and insert of linear dependence.

## Data Availability

The data that support the findings of this study are available on request from the corresponding author. The data are not publicly available due to privacy or ethical restrictions.

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
