# Peer review of "Submicronic-Scale Mechanochemical Characterization of Oxygen-Enriched Materials"

_nanomaterials, 2024, doi:10.3390/nano14070628_

Round 1

Reviewer 1 Report

Comments and Suggestions for Authors

The manuscript titled “Submicronic-scale mechanochemical characterization of oxygen enriched materials” by Garnier, M.; et al. is a scientific work where the authors studied the relationship between the elastic deformation of oxygen enriched titanium alloys and their microwave signal. This proof-of-concept research could be extandable for other metallic materials which have a strong impact in many industrial sectors. The manuscript is generally well-written and this is a topic of growing interest.

However, it exists some points that need to be addressed (please, see them below detailed point-by-point) to improve the scientifc quality of the submitted manuscript paper before this article will be consider for its publication in Nanomaterials.

1) KEYWORDS. The authors should consider to add the term related to the chemical nature (oxide-metal interface materials) of the examined samples in the keyword list.

2) INTRODUCTION. “A major problem in the metallurgy (…) conditions that have an impact on their lifespan” (lines 26-28). Could the authors provide quantitative details about the lifespan decrease related to these severe conditions?

3) “It is therefore crucial to characterize this oxide layer (…) understand the impact of oxygen diffusion on the material’s mechanical properties” (lines 55-58). Here, even if I agree with this statement provided by the authors, it may be advisable to remark the importance of the local nanomechanical properties [1] which could significantly impact on the material properties like the case of the microstructure, electrical or corrosive, among others [2].

[1] Magazzù, A.; et al. Investigation of Soft Matter Nanomechanics by Atomic Force Microscopy and Optical Tweezers: A Comprehensive Review. Nanomaterials 2023, 13, 963. https://doi.org/10.3390/nano13060963.

[2] Dhiflaoui, H.; et al. Influence of TiO2 on the Microstructure, Mechanical Properties and Corrosion Resistance of Hydroxyapatite HaP + TiO2 Nanocomposites Deposited Using Spray Pyrolysis. Coatings 2023, 13, 1283. https://doi.org/10.3390/coatings13071283.

4) “Energy Dispersive X-ray Spectroscopy (EDS), and Nuclear Reaction Analysis (NRA) (…) Energy dispersive X-ray spectroscopy (EDS) techniques (…) Nuclear reaction analysis (NRA) (…)” (lines 62-72). The full-names of all the terms should be defined with their respective abbreviations the first time that they appear in the main mnuscript body text. It is not neccesary set them twice.

5) THEORETICAL APPROACH. “In the scanning microwave microscope, the contact mode is often used with minimial interaction force, typically less than 750 pN” (lines 129-130). How is the strategy used by the authors to minimize the tip-sample capillar forces existing in air conditions. A brief statement should be provided in this regard.

6) EXPERIMENTS AND RESULTS. “To characterize the residual stresses in the material, an incremental hole-drilling method was used (…) The process entails drilling a hole at the desired measurement location, leading to the release of locked-up residual stresses” (lines 177-182). What was the methodology used by the authors to drill the holes? A brief statement should be provided in this regard. Indeed, the authors should add all the software tools used for the different type of measurements devoted in this work.

7) “3.2.3. Nanoindentation measurements” (lines 345-363). A representative indentation curve should be added in the Figure 8 (line 335). What was the physical model used by the authors to estimate the Young’s modulus?

8) DISCUSSION. This section clearly depicts the gathered data in this research with feasible explanations related to the nature of the tested materials. No actions are requested from the authors.

9) CONCLUSIONS. This section perfectly remarks the most relevant outcomes found by the authors in this field. The authors should add a brief statement to discuss about the future line actions to pursue this research and the open perspectives.

Comments on the Quality of English Language

The manuscript is generally well-written albeit it may be desirable if the authors could recheck it in order to polish final details susceptible to be improved.

Author Response

Thank you very much for taking the time to review this manuscript. Please find the detailed responses below and the corresponding corrections highlighted in the re-submitted files.

1) Thank you for your suggestion. We added the keyword " oxide-metal interface materials " (line 23).

2) A lifespan is impacted by a wide diversity of “severe conditions”. Indeed, the severe conditions can be high and cyclic loads, high temperatures, corrosive atmospheres (pH, salt, chlorides, sulfides, fluorides, etc.), hardening atmospheres (O2, H2, N2), irradiations. Thus, what is generally called “severe conditions” are precisely the exogen parameters that decrease the lifespan of industrial parts. Thus, the impact on lifespan is observable, but general quantification of such impact is impossible.

3) References were taken into consideration (lines 58-60).

4) The full names have been replaced by their respective abbreviations (lines 70 and 74).

5) To minimize capillary forces, the experimental device is maintained below 30% RH. This is the level at which capillary forces are minimal. The temperature is below 25°C.

6) The incremental hole drilling method is a well-known and quite old method for residual stress quantifications. The international standard ASTM E837 is devoted to this characterization method. We have added the reference to this standard within the text of the article, as well as a short description of the method (lines 179-185). 

7) The load-discharge curve for an indent has been added in figure 8 (a) (line 345). The model used to estimate the Young’s module has been developed in the manuscript (lines 323-338).

9) Future actions and perspectives have been added in the conclusion (lines 445-448). The next stage of our work will involve studying other types of metallic material and different levels of oxidation to calibrate the device (both in phase and amplitude).

Reviewer 2 Report

Comments and Suggestions for Authors

The authors of this manuscript present an  innovative approach for high resolution characterization of oxygen enrichment in a Ti-based alloy.

However several details should be addressed before publication:

1. line 156,296,301,319 several typo should be corrected

2. In Fig. 6 and Fig. 7 was written F= 7.57 GHz, and F = 7.75 GHz, possible typo! If not, explication should be provided why are two values for F.

3. A surface of  80 x 80 um^2 for investigation is considered but in Fig. 6 d/e the variation on Ox is longer than 80 um. Explication should be provided why?

4. line 386, signification of R^2 ?

5. Was tested the method on other type of alloys ?

6. Was tested the method on same alloy but with different level of oxidation ? If Yes, results of a comparative investigation should be provided.

Comments on the Quality of English Language

Minor editing of English language required

Author Response

Thank you very much for taking the time to review this manuscript. Please find the detailed responses below and the corresponding corrections highlighted in the re-submitted files.

1. The errors have been corrected.

2. The typing error has been corrected.

3. We have reworded the sentence explaining the profiles obtained in figure 6d and 6e, which are profiles obtained by two adjacent scans of 80 micrometers each (Line 303-304).

4. R² is Pearson's coefficient of linear determination. It is a measure of the predictive quality of a linear regression. The signification of R² has been defined (line 397).

5. and 6. Work is in progress on other types of metal samples, as well as on samples with different levels of oxidation. This work will enable the measurement technique to be calibrated.